# Peroxisomal PEX7 Receptor Affects Cadmium-Induced ROS and Auxin Homeostasis in Arabidopsis Root System

**DOI:** 10.3390/antiox10091494

**Published:** 2021-09-20

**Authors:** Diego Piacentini, Federica Della Rovere, Ilaria Bertoldi, Lorenzo Massimi, Adriano Sofo, Maria Maddalena Altamura, Giuseppina Falasca

**Affiliations:** 1Department of Environmental Biology, Sapienza University of Rome, Piazzale Aldo Moro 5, 00185 Rome, Italy; diego.piacentini@uniroma1.it (D.P.); federica.dellarovere@uniroma1.it (F.D.R.); ilaria.bertoldi.95@gmail.com (I.B.); l.massimi@uniroma1.it (L.M.); mariamaddalena.altamura@uniroma1.it (M.M.A.); 2Department of European and Mediterranean Cultures: Architecture, Environment, and Cultural Heritage (DICEM), University of Basilicata, Via San Rocco 3, 75100 Matera, Italy; adriano.sofo@unibas.it

**Keywords:** *Arabidopsis thaliana*, cadmium, H_2_O_2_, oxidative stress, peroxisomal targeting signal, peroxisome, PEX7/Peroxin7 receptor, root system

## Abstract

Peroxisomes are important in plant physiological functions and stress responses. Through the production of reactive oxygen and nitrogen species (ROS and RNS), and antioxidant defense enzymes, peroxisomes control cellular redox homeostasis. Peroxin (PEX) proteins, such as PEX7 and PEX5, recognize peroxisome targeting signals (PTS1/PTS2) important for transporting proteins from cytosol to peroxisomal matrix. *pex7-1* mutant displays reduced PTS2 protein import and altered peroxisomal metabolism. In this research we analyzed the role of PEX7 in the *Arabidopsis thaliana* root system exposed to 30 or 60 μM CdSO_4_. Cd uptake and translocation, indole-3-acetic acid (IAA) and indole-3-butyric acid (IBA) levels, and reactive oxygen species (ROS) and reactive nitrogen species (RNS) levels and catalase activity were analyzed in *pex7-1* mutant primary and lateral roots in comparison with the wild type (wt). The peroxisomal defect due to *PEX7* mutation did not reduce Cd-uptake but reduced its translocation to the shoot and the root cell peroxisomal signal detected by 8-(4-Nitrophenyl) Bodipy (N-BODIPY) probe. The trend of nitric oxide (NO) and peroxynitrite in *pex7-1* roots, exposed/not exposed to Cd, was as in wt, with the higher Cd-concentration inducing higher levels of these RNS. By contrast, *PEX7* mutation caused changes in Cd-induced hydrogen peroxide (H_2_O_2_) and superoxide anion (O_2_^●−^) levels in the roots, delaying ROS-scavenging. Results show that PEX7 is involved in counteracting Cd toxicity in Arabidopsis root system by controlling ROS metabolism and affecting auxin levels. These results add further information to the important role of peroxisomes in plant responses to Cd.

## 1. Introduction

Peroxisomes play important roles in a wide range of plant physiological functions, including primary and secondary metabolism, development, and stress responses. In the latter responses, they work through the production of reactive oxygen species (ROS), reactive nitrogen species (RNS), and antioxidant defense enzymes [1,2]. For example, these organelles have an oxidative metabolism characterized by high levels of hydrogen peroxide (H_2_O_2_), as a ROS, but also by the presence of catalase (CAT), as an H_2_O_2_-scavenging enzyme [3]. Being highly dynamic structures, peroxisomes respond to environmental and cellular cues by changing their size, number, and proteomic content [1,4]. The proteomic content includes at least 200 proteins [5], imported from the cytosol to maintain and modulate peroxisomal functions [6]. Peroxisomes are involved in lipid mobilization through β-oxidation, nitrogen metabolism, synthesis, and metabolism of plant hormones [7]. For example, the β-oxidation of the auxin precursor indole-3-butyric acid (IBA) into the active auxin indole-3-acetic acid (IAA) occurs in the peroxisomes [8], even if how/whether the auxin pathways are related to stress response and to defense compounds produced in the peroxisomes still needs investigation.

To maintain their functions/metabolism, the peroxisomes use two types of targeting signals for matrix import of soluble proteins from the cytosol, i.e., peroxisome-targeting signals—PTS1, a C-terminal tripeptide, and—PTS2, an N-terminal nonapeptide import [6,9]. The matrix protein import requires protein-named peroxins (PEX), encoded by *PEX* genes [10,11] and that recognize PTS. PTS1 and PTS2 are recognized by the receptors PEX5 and PEX7, respectively [11]. PEX5 and PEX7 form a complex, with PEX5 requiring PEX7 for stability, and PEX7 requiring PEX5 for cargo delivery, and after the delivery, both receptors are returned to the cytosol for further peroxisomal protein import, even if PEX7 recycling is still widely unexplored [11]. Moreover, some PEX proteins, such as the peroxisomal membrane PEX11, are involved in the control of peroxisome proliferation in different plant species exposed to stressful conditions such as heavy metals [12] or salinity [13]. However, a similar role for the peroxisomal matrix proteins, such as PEX7, is still unknown. An *Arabidopsis thaliana* peroxin mutant, named *pex7-1*, showing peroxisome-defective phenotype including reduced PTS2 protein import, has been characterized demonstrating that *Arabidopsis thaliana* PEX7 is necessary for PTS2 protein peroxisomal import and that its function is relevant for the composition of peroxisomal protein content, and blocked in *pex7-1* mutant [14]. Even if *pex7-1* mutant has been demonstrated to be resistant to IBA [15], and an inefficient β-oxidation of IBA into IAA is defective in PTS2 proteins targeting [16], the importance of PEX7 in PTS2-protein import into the peroxisome and the relationship with IBA metabolism, even if suggested [14], remains to be investigated, in particular in plants that must mitigate the toxicity of metals present in the soil.

Peroxisomal proteome analyses have been performed on various organs of *Arabidopsis thaliana*, including cotyledons [17] and leaves [18]. However, the roots have not yet been analyzed [2]. By contrast, the root system is important because the development of the shoot depends on the correct development and functioning of the roots. Each component of the root system, i.e., primary root (PR), lateral roots (LRs), and adventitious roots (ARs) are under the developmental control of auxin, with IAA as the main root-inducer. In plant, the auxin pool comprises free IAA, IAA conjugates, and the IAA-precursor IBA [19]. In *Arabidopsis thaliana*, IBA activity completely depends on its conversion into IAA in the target cells [20,21]; however its formation occurs by a modulated conversion into IAA involving nitric oxide (NO) production, as demonstrated for AR formation [22,23,24].

The root system is the first part of the plant to contact, and react to, stress agents in the soil, for example pollutants such as Cadmium (Cd). Cadmium is easily absorbed by the roots and its toxicity affects the root growth, leading to a precocious tissue differentiation, to oxidative stress and even to cell death [25]. In *Arabidopsis thaliana*, the exposure to specific concentrations of CdSO_4_ (30 and mainly 60 µM) alters the elongation and primary tissue organization of the PR and LRs, changing the root system architecture [26]. In the same plant, the exposure to specific Cd concentrations inhibits the PR growth by affecting the stem cells in the root apex [27]. The same pollutant negatively affects stem cell identity also in the LR and AR apices, by affecting auxin localization and levels [25].

Cadmium exerts toxicity mainly by inducing oxidative stress in a ROS-dependent manner [12], and through an imbalance between the production of ROS and RNS, and their detoxification [28]. In accordance, in rice root system, Cd decreases the endogenous NO-content, increases H_2_O_2_ formation, and alters biosynthesis and levels and distribution of auxin [29]. However, exogenous treatments of IAA, but mainly IBA, combined with Cd, mitigate the pollutant effects on the roots [29]. Peroxisomes contribute to the cellular redox homeostasis by controlling the levels not only of ROS, especially superoxide anion (O_2_^●−^) and H_2_O_2_, but also of RNS, especially NO and its derived molecule peroxynitrite (ONOO^−^), whose presence in peroxisomes has been demonstrated [30]. Being the product of the reaction of NO with O_2_^●^^−^, the ONOO^−^ is an example of ROS/RNS crossroad.

It has been recently demonstrated that the ROS/RNS crosstalk depends on the molecules involved and, on their concentrations, is organelle- and microcompartment-specific, and might have beneficial or deleterious effects on plant cells [31]. However, it is important to highlight that the endogenously generated ROS have also a morphogenic role in the roots, by regulating the balance of cell proliferation and differentiation [32,33,34]. Under normal conditions, ROS concentration in peroxisomes is controlled; however, peroxisomal ROS homeostasis is disrupted in the presence of Cd, and RNS are generated [5,35,36]. The NO may act as a signaling molecule coordinating development and stress responses, but also as an oxidative stress inducer [36]. Recently, differences in the peroxisomal response to Cd have been observed between the PR and the LRs of *Arabidopsis thaliana* because the pollutant causes significant changes in peroxisome distribution, size, and NO content mainly in the PR apex [37]. It has also been demonstrated that NO can modulate the levels of auxin, by affecting its synthesis, transport, signaling, and degradation [38].

It has been hypothesized that under Cd stress, a modulation of NO occurs for controlling H_2_O_2_ levels [39], with a possible positive relationship with PEX7 cargo delivery to the peroxisomal matrix. It is known that H_2_O_2_ can act as a stress transducer to reduce or improve plant stress reactions [40,41], and that it acts synergistically or antagonistically with plant growth regulators such as auxins, and with signaling molecules, such as NO, under a lot of environmental stresses including Cd stress [42]. In *Arabidopsis thaliana*, transcriptomic analyses of the *cat2* mutant, deficient in CAT activity to reduce H_2_O_2_ levels, demonstrate that H_2_O_2_ produced by peroxisomes induces prevailing protein repair responses [43]. However, the mitigative role of H_2_O_2_ on Cd stress differs based on plant species and H_2_O_2_ concentrations [44], and even may not occur. In fact, in *Brassica rapa*, belonging to the same family of *Arabidopsis thaliana*, exposure at high concentration of Cd results into increases in both H_2_O_2_ and O_2_^●−^ in the root tips, leading to oxidative injury followed by root growth inhibition [45]. Whether Cd stress affects root growth either by differentially regulating endogenous H_2_O_2_ and O_2_^●−^, or by inducing oxidative injury, remains to be determined [45]. In *Arabidopsis thaliana*, the PR elongation is positively regulated by O_2_^●−^ in the elongation/distention zone, while negatively by H_2_O_2_ in the differentiation zone [46]. In accordance, Cd stimulates the production of H_2_O_2_ and inhibits that of O_2_^●−^ in the roots of *Glycine max* and *Cucumis sativus* [47]. By contrast, both H_2_O_2_ and O_2_^●−^ are indispensable for the emergence of LRs in *Arabidopsis thaliana* [48]. However, whether and how ROS act as signaling molecules rather than as oxidative stress inducers to regulate PR/LR growth under Cd exposure remains obscure for *Arabidopsis thaliana*.

In plants, CATs are the most abundant peroxisomal ROS antioxidant enzymes [39,43,49,50,51]. Superoxide dismutase (SOD) and CATs scavenge ROS by converting superoxide to H_2_O_2_ and H_2_O_2_ to oxygen and water, sequentially. An increase in CAT activity helps plants to react to heavy metal stress [52,53], although a decrease in its activity under some stress conditions has also been reported [37]. Catalases do not require additional reductants to eliminate H_2_O_2_ [54]. *Arabidopsis thaliana* genome contains three *CAT* genes (*CAT1*, *CAT2*, and *CAT3*) [39]. The CAT proteins are important under unfavorable conditions for plants. For example, CAT1 is implicated in the drought and salt stress responses [55], CAT3 participates in the drought stress response [56], and CAT2 is involved in plant response to heat, heavy metal [57,58], cold, and salt stresses [59], and is the main CAT to degrade H_2_O_2_ in peroxisomes [60,61]. During plant development, CATs exert morphogenic roles being involved in many processes, including root growth [31,62].

Altogether, the peroxisomal responses to Cd of the *Arabidopsis thaliana* root system still need to be elucidated and are important given the importance of the roots as the first plant organs to become in contact with the pollutant in the soil.

The purpose of this research was to shed light on the mechanisms of action of peroxisomes in plant responses to cadmium toxicity and, in particular, to verify whether peroxisomal proteins, such as PEX7, were involved in this process. In this regard, an increase in the knowledge of the role of peroxisomes in abiotic stress responses is required for the emerging potential of these organelles in the field of green biotechnology, which could represent a promising strategy for increasing stress tolerance of economically important crops in the future [63].

To this aim, the *Arabidopsis thaliana* peroxin mutant *pex7-1*, which displays a reduced PTS2 protein import into peroxisomal matrix, was used to identify the peroxisome roles in the root system development after Cd exposure. Emphasis is placed on understanding how these organelles work in PR and LR reaction and protection to the pollutant, and how/whether PEX7 receptor activity is involved. Cadmium absorption, changes in IAA content, in the conversion of IBA into IAA, in NO and ONOO^−^ levels, in superoxide radical formation, and in H_2_O_2_ production and elimination by CAT scavenging activity, were investigated in the presence/absence of PEX7 receptor activity and the pollutant.

Results show that PEX7 is involved in Cd translocation from roots to shoots. The peroxisomal root cell fluorescence signal is enhanced in response to the pollutant in a concentration dependent manner, with the PTS2-depending protein import into the peroxisomal matrix involved. Both PR elongation and LR formation and elongation are negatively affected by Cd, and the IBA-to-IAA conversion is reduced in the presence of *pex7-1* mutation, with this altering the auxin balance in the mutant roots. PEX7 activity is not involved in changing RNS levels in response to Cd, whereas it is positively involved in changing ROS levels and in accelerating the CAT scavenging activity. Altogether, results demonstrate that well-functioning peroxisomes are indispensable to the *Arabidopsis thaliana* root system for reacting to Cd, and that a correct import of PTS2-proteins by PEX7 into the peroxisomal matrix is essential for ROS scavenging action, and for controlling auxin homeostasis.

## 2. Materials and Methods

### 2.1. Plant Material and Growth Conditions

Seeds of *A. thaliana* (L.) Heynh ecotype Columbia (Col-0, wt), and of *pex7-1* mutant line [14] were surface-sterilized for 5 min in a solution of 70% (*v*/*v*) ethanol and 0.1% (*w*/*v*) SDS and then placed for 20 min in water containing 20% (*v*/*v*) bleach and 0.1% SDS. Then, the seeds were washed four times in sterile water and sown on half-strength Murashige and Skoog medium [64] (Duchefa Biochemie), with 0.7% (*w*/*v*) sucrose, 0.7% (*w*/*v*) agar and with/without either 30 or 60 μM CdSO_4_ (Sigma-Aldrich, St. Louis, MO, USA) at pH 5.6–5.8. Then, the plates containing the seeds were vertically incubated under long-day conditions (16 h light/8 h dark) at 22 ± 2 °C and 100 μE m^−2^s^−1^ of light intensity for 10 days. The two CdSO_4_ concentrations were selected based on the results of our previously published data [26,65].

### 2.2. pex7-1 Mutant Screening Procedure

Seeds of *A. thaliana* mutant *pex7-1* (*PEX7* gene accession number: *At1g29260*), isolated from the wt T-DNA insertion line SALK_005354, were kindly provided by Prof. Bonnie Bartel (Rice University, Houston, TX, USA). Total DNA isolation was performed from wt and *pex7-1* rosette leaves by using the DNeasy Plant Mini Kit (QIAGEN, Hilden, Germany). According to Woodward and Bartel [14], amplification with the oligonucleotide PEX7–2 (5′-CTTCTCGAAGATTCAATTCAACGAT-3′) and the modified LBb1 T-DNA left border primer LB1-Salk (5′-CAAACCAGCGTGGACCGCTTGCTGCA-3′) yielded a ~200-base pair product from mutant DNA, whereas PEX7 (5′CTCGAATTTAGATTTCTCTCTCACTTTTA-3′) combined with PEX7–2 yielded a 252-base pair product in the presence of wt DNA, enabling genotypic determination. PCR amplifications of 35 cycles (94 °C for 30 s, 60 °C for 30 s, 72 °C for 5 min) were carried out in volumes of 25 µL containing 0.5 µM of each primer, 2.5 mM of MgCl_2_ solution, 5 mM of dNTP Mix, 0.5 × NH_4_ reaction buffer, 0.25 units of BioTaq (Bioline, London, England), and 3 µL of DNA template. Then, 5 µL of PCR products were run in 1.5% (*w*/*v*) agarose gel stained with GelRed (Biotium, Fremont, CA, USA) and gel images were scanned with Gel Doc XR+ (Bio-Rad, Hercules, CA, USA). Homozygous T1 seeds were collected and used for the experiments.

### 2.3. Root and Shoot Cd Determination by ICP-OES

For Cd content analysis, wt and *pex7-1* seedlings grown in the presence or not of 30 or 60 µM CdSO_4_ were collected and washed with an ice-cold 5 mM CaCl_2_ solution for 10 min to displace extracellular Cd. Then, the seedlings were separated into roots and shoots, oven-dried at 70 °C for 72 h and weighed. A total of 0.1 g DW of each sample was subjected to a microwave assisted acid digestion for 30 min at 180 °C by using a HNO_3_/H_2_O_2_ mixture (2:1, *v*/*v*). The digested solutions were then diluted to 100 mL with Milli-Q water and filtered with syringe filters (25 mm in diameter, 0.45 μm pore size). Cadmium concentrations were determined by inductively coupled plasma optical emission spectrometer (ICP-OES; Vista MPX CCD Simultaneous; Varian, Victoria, Mulgrave, Australia) using axial-view mode and equipped with a cyclonic spray chamber. Standard solutions for daily six-point calibration were matrix-matched by preparation in 0.01 M CH_3_COOH/CH_3_COOK solution (pH 4.5) or HNO_3_ 10% (*w*/*w*). To control nebulizer efficiency, an internal standard (yttrium 100 μg L^−1^, wavelength = 371.030 nm) was used. External standard calibration curve was performed for Cd by serially diluting standard stock solution (1000 ± 2 mg L^−1^, Exaxol Italia Chemical Manufacturers Srl., Genoa, Italy). The values of blanks, subjected to similar sample preparation and analytical procedures, were deducted from all measurements and the limits of detection were set at 3 times the standard deviation (SD) of 10 replicate blank determinations. Standard deviations of the replicates were all below 8%. The instrumental conditions and the performance of the method were according to Canepari and co-workers [66]. The obtained Cd contents were divided by the dry weight of each sample to obtain mg/g concentrations. The translocation factor (TF) was calculated as the ratio of the total heavy metal concentration in the shoot to that in the root [67].

### 2.4. Root Morphological Analysis

Primary root (PR) length from 30 seedlings per genotype and treatment was measured under a LEICA MZ8 stereomicroscope using Zeiss Zen 2.3 software from digital images (Zeiss, Oberkochen, Germany), captured with Zeiss AxioCam camera (Zeiss, Oberkochen, Germany) and expressed as mean length (cm ± SE). Lateral roots (LRs) were counted under a Leica DMRB microscope and the corresponding LR density was expressed as mean number of LRs per cm of PR. In addition, the ratio between the number of mature LRs and of lateral root primordia (LRPs) produced in each PR was determined and expressed as a mean value.

### 2.5. Auxin Quantification

For IBA and IAA quantification, wt and *pex7-1* seedlings non-exposed (Control) or exposed to 30 or 60 μM CdSO_4_ were collected, separated into roots and shoots, and 0.6 g FW of each sample was immediately stored at −80 °C until use. The extraction of IAA and IBA was carried out on 300 mg-aliquots grounded into powder with liquid nitrogen. To each sample, 3.0 mL of extraction solvent (2-propanol/H_2_O/HCl 37%; 2:1:0.002, *v*/*v*/*v*) was added. The tubes were shaken at a speed of 13,000× *g* for 20 min at 4 °C. To each tube, 3.0 mL of dichloromethane was added. Then, the samples were shaken for 30 min at 4 °C and centrifuged at 15,000× *g* for 5 min. After centrifugation, 1.0 mL of the solvent from the lower phase was transferred into a screw-cap vial, and the solvent mixture was concentrated using an evaporator with nitrogen flow. Finally, the samples were re-dissolved in 60 mL methanol and stored at −20 °C before quantitative analysis. The quantitative determinations of IAA and IBA were carried out by high-performance liquid chromatography coupled with mass spectrometry, according to Veloccia and co-workers [22]. Pure standards of IAA and IBA were used for quantification (Duchefa Biochemie, Haarlem, The Netherlands). The internal standards used were [^2^H_5_] IAA and [^2^H_9_] IBA (OlChemIm Ltd., Olomouc, Czechia; crystalline form, purity > 97% for HPLC). The IBA and IAA contents in the shoots of both the genotypes for all treatments are provided in Appendix A.

### 2.6. Root Protoplast Isolation and N-BODIPY Fluorescence Assay

Protoplasts were isolated from 0.5 g of the root system of wt and *pex7-1* seedlings grown in the absence (Control) or in the presence of 30 or 60 µM CdSO_4_ by following the enzymatic digestion procedure of Lindberg and co-workers [68]. The same number of protoplasts from wt and *pex7-1* were loaded with 1 µM of 8-(4-Nitrophenyl) Bodipy (N-BODIPY, Santa Cruz Biotechnology) in dimethylsulphoxide (DMSO, Sigma-Aldrich, St. Louis, MO, USA) for 2 h in complete darkness and at RT. The employment of N-BODIPY as peroxisomal marker was justified by its high specificity to these organelles, thus allowing to image peroxisomes by fluorescence microscopy, even under abiotic stress conditions [69]. The fluorescence signal was observed using a DMRB microscope equipped with a 450–490 nm excitation/515–565 nm emission filter set. For each root protoplast observed, 4 sections at different depth levels along the z-axis were acquired with a LEICA DC500 digital camera (Leica, Wetzlar, Germany) and processed with Zerene Stacker software (version 1.04; Zerene Systems, Richland, WA, USA). The quantification of the N-BODIPY signal was performed among the protoplasts of the same size (~200 µm^2^) using ImageJ software (version 1.53c, Wayne Rasband, National Institutes of Health, Bethesda, USA. Available online: https://imagej.nih.gov/ij, accessed on 1 June 2021) and expressed in arbitrary units (A.U.s; from 0 to 255). The same microscope and image acquisition settings of the fluorescence signals were used for the whole experiment.

### 2.7. In Situ Root ROS and RNS Visualization

The PRs and LRs of 30 seedlings per genotype and treatment were analyzed for in vivo hydrogen peroxide (H_2_O_2_), nitric oxide (NO), peroxynitrite (ONOO^−^), and superoxide anion (O_2_^●−^) visualization. H_2_O_2_ was detected after 3, 4, and 10 days after germination using 2′,7′-dichlorofluorescin diacetate (DCFH-DA, Sigma-Aldrich, St. Louis, MO, USA) and after 10 days after germination by 3,3′-diaminobenzidine tetrahydrochloride hydrate (DAB, AppliChem GmbH). NO, ONOO^−^, and O_2_^●−^ were detected using 4-amino-5-methylamino-2′,7′-difluorofluorescein diacetate (DAF-FM DA, Invitrogen^TM,^ Waltham, MA, USA), 3′-(p-aminophenyl) fluorescein (APF, Invitrogen^TM,^ Waltham, MA, USA), and nitro blue tetrazolium (NBT, Roche Diagnostics Corp., Basel, Switzerland), respectively, according to Piacentini and co-workers [28,70]. For DAB staining, the samples were incubated in 1 mg/mL DAB-HCl dissolved in milliQ water for 1 h in complete darkness and at RT, after 5 min of vacuum infiltration. The pH of the solution was set to 7.0 with 200 mM of Na_2_HPO_4_. For DCFH-DA, DAF-FM DA, and APF analyses, the samples were stained with 10 µM of the specific fluorescent probe in 10 mM Tris-HCl buffer (pH 7.4) for DCFH-DA and APF or in 20 mM HEPES/NaOH for DAF-FM DA. The incubation period was 30 min (DCFH-DA and DAF-FM DA) or 1 h (APF) at 25 °C in darkness. The study of the fluorescence signals was performed with a Leica DMRB epifluorescence microscope equipped with a LEICA DC 500 camera and with a 450–490 nm excitation/515–565 nm emission filter set. The fluorescence intensity was quantified on PRs using ImageJ software (version 1.53c, Wayne Rasband, National Institutes of Health, Bethesda, USA. Available online: https://imagej.nih.gov/ij, accessed on 1 June 2021) and expressed in arbitrary units (A.U.s; from 0 to 255). The fluorescence signal in LRs was not quantified due to the low number of elongated LRs. The same microscope and image acquisition settings were used for the whole experiment depending on the fluorescent probe used. Finally, DAB and NBT-stained roots were kept in the chloral hydrate solution (Sigma-Aldrich, St. Louis, MO, USA) before visualization and then observed under white light with the above-mentioned microscope equipped with Nomarski optics. Images of wt and *pex7-1* root systems, after NBT staining, were also acquired with an Axio Imager M2 microscope (Zeiss, Oberkochen, Germany) motorized on the 3 axes. Image tiles scanning were performed with an Axiocam 105 (Zeiss, Oberkochen, Germany) camera using Zen pro 2.5 (Zeiss, Oberkochen, Germany) software.

### 2.8. Catalase (CAT) Activity Assays

For the detection of CAT activity, the root system of wt and *pex7-1* seedlings grown in the presence or not of 30 or 60 μM CdSO_4_ was collected and ground with liquid nitrogen in pre-chilled mortars and pestles. The powder (0.1 g of each sample) was suspended in a 50 mM Tris-HCl (pH 7.8, ratio 1:4; *w*/*v*) buffer containing 100 μM EDTA, 0.2% (*v*/*v*), Triton X-100 and 10% (*v*/*v*) glycerol. The homogenates were filtered and centrifuged at 13,300 rpm for 20 min and the supernatants were collected and used for the enzymatic assay. The activity of CAT (EC 1.11.1.6) was spectrophotometrically (Beckman DU-530, Beckman Counter, Inc., Fullerton, CA, USA) estimated according to the method of Aebi [71], by following the disappearance of H_2_O_2_ at 240 nm for 2 min. The final volume of the reaction mixture was 1 mL containing 50 mM phosphate buffer (pH 7.0), 10.6 mM H_2_O_2_, and 30 µL of root extract. Protein content was determined at 595 nm using the Bio-Rad protein assay (Bio-Rad, Hercules, CA, USA), using a bovine serum albumin (BSA, Sigma-Aldrich, St. Louis, MO, USA) solution to prepare the standard curve.

### 2.9. Statistical Analysis

All the data were statistically analyzed using one-way or two-way ANOVA test followed by Tukey’s post-test (at least at *p* < 0.05) after performing both Shapiro–Wilk’s test and Bartlett’s test. The statistical analyses were carried out through GraphPad Prism (Version 8.0.2; GraphPad Software, San Diego, CA, USA) and RStudio (Version 1.2.5042; RStudio, Boston, MA, USA) software. All the experiments were performed in three independent biological replicates with similar results. In most of the figures, data are the mean of the three biological replicates, otherwise data are from the first biological replicate.

## 3. Results

### 3.1. pex7-1 Mutation Reduces Cd Translocation from Root to Shoot

To investigate whether *pex7-1* mutation affects Cd up-take and translocation, the Cd content was evaluated in the root system and in the shoot of wt and *pex7-1* plants. In both genotypes, Cd accumulated mainly in the roots, and in a dose-dependent manner (Figure 1a). However, when exposed to the higher Cd concentration, a greater and significant accumulation of the heavy metal was observed in *pex7-1* roots compared to wt roots (Figure 1a). In both the genotypes, the transport of Cd to the shoot was low; however, it was significantly reduced in the mutant in comparison with wt, independently from the Cd concentration used (Figure 1b).

Indeed, the evaluation of the translocation factor (TF) showed that the *pex7-1* mutant had a significantly (*p* < 0.01 difference for both Cd concentrations) reduced capability to translocate Cd from the root to the shoot in comparison with wt. In fact, in the mutant, the TF in the presence of 30 and 60 µM Cd was 0.33 and 0.39, respectively, whereas 0.63 and 0.54 in wt (Figure 1c). This highlights a positive role of PEX7 in the Cd root-to-shoot translocation in *Arabidopsis thaliana*.

### 3.2. pex7-1 Mutation Reduces LR Elongation in Cd Presence

The effects of Cd on the root system of wt and *pex7-1* mutant were evaluated. The PR length of the mutant plants was significantly reduced in the presence of Cd, in a concentration-dependent manner, the same as in the wt (Figure 2a,d). Lateral root density was also reduced in both the genotypes in the presence of Cd (Figure 2b). In the presence of 60 µM Cd, despite a greater Cd accumulation in the mutant roots (Figure 1a), a similar reduction in LR density was observed in the mutant in comparison with the wt (Figure 2b). The few LRs formed in the mutant in the presence of the higher Cd exposure remained mainly at the primordium stage, on the contrary, a greater number of LRs in the wt were elongated (Figure 2c). This can be because of a more altered auxin balance induced by the pollutant in the mutant roots in comparison with the wt.

### 3.3. pex7-1 Mutation Reduces IBA to IAA Conversion

To verify whether the defect in the conversion of IBA into IAA of the *pex7-1* mutant played a role in the plant response to Cd stress, and whether it was responsible for the morphological root responses (Figure 2), the levels of the two main auxins were detected in the root system and in the aerial organs of wt and the mutant after exposure or not to Cd. Cadmium, at both concentrations, induced a significant (*p* < 0.01) increase in the IAA levels in comparison with Control in both roots and shoots of wt and *pex7-1* (Figure 3a, Appendix A). The levels of IBA were, as expected, lower than those of IAA in roots and shoots of both genotypes. However, *pex7-1* roots showed significantly higher IBA levels than wt already in the Control, and they further increased in the presence of Cd (Figure 3b). The IBA levels in the shoots significantly increased in the presence of Cd, compared to the Control, and similarly in the wt and mutant (Appendix A). This highlights that the mutant’s reduced ability to convert IBA into IAA is accentuated in the presence of Cd toxicity mainly in the root system.

### 3.4. pex7-1 Mutation Reduces Cd-Induced Root Peroxisomal Signal

To deepen the role of peroxisomes in the responses to Cd toxicity, the variations of the peroxisomal signal after the heavy metal exposure were analyzed in *pex7-1* root protoplasts in comparison with the wt (Figure 4) by the use of the N-BODIPY probe [69]. In Cd absence, a higher (Figure 4a,d,g,h) and more diffuse (Figure 4a,d) peroxisomal signal was observed in *pex7-1* protoplasts in comparison with the wt. Cadmium treatments induced an increase in the peroxisomal signal in the protoplasts extracted from wt roots (Figure 4b,c), which at the higher Cd concentration became significantly higher than the Control treatment (Figure 4g). Interestingly, the protoplasts extracted from the roots of the mutant showed a progressive decrease in the peroxisomal signal in the presence of Cd (Figure 4e,f) in comparison with the Control (Figure 4d). Indeed, the peroxisomal signal was strongly reduced at the higher Cd concentration in comparison with the Control (Figure 4f,h). These results demonstrate that the peroxisomes of *pex7-1*, already altered by the mutation, are more damaged by the presence of the heavy metal than those of the wt.

### 3.5. Cd-Induced Nitric Oxide and Peroxynitrite Levels Do Not Change in pex7-1 Roots Compared to wt

The level of NO in the wt and *pex7-1* PRs exposed or not to the two Cd concentrations was monitored, using DAF-FM DA probe, to verify whether the alteration in the peroxisomal receptor PEX7 modified the NO levels in the root cells exposed to Cd stress. The trend of NO levels in the wt showed a very low NO signal in the roots under the Control and 30 µM Cd (Figure 5a,b,g), which was mainly localized in the cortical parenchyma in the absence of the pollutant (Figure 5a). However, the signal significantly (*p* < 0.01) increased in the presence of 60 µM Cd (Figure 5g), showing also strong cellular accumulations in all the tissues of the organ, especially those close to the meristematic zone (Figure 5c). The levels and the distribution of NO in the mutant were similar to those of the wt under all treatments (Figure 5d–f,h), with a significant increase after the treatment with the higher Cd concentration also in this case (Figure 5h).

The detection of ONOO^−^ levels, through APF probe, in the PRs of the wt in comparison with the mutant showed that the mutation did not affect the response of this RNS. In fact, ONOO^−^ fluorescence signal was similarly low in *pex7-1* and the wt roots of the Control and 30 µM Cd treatments, and with the same distribution in both the genotypes (Figure 6a,b,d,e). In addition, the signal significantly (*p* < 0.01) increased in both genotypes in the presence of 60 µM Cd, spreading in all the tissues of the organ except for the root apex (Figure 6c,f–h). Altogether, these results suggest that the mutation does not affect the production/localization of these two RNS in the roots, even during Cd-related stress conditions.

### 3.6. Cd-Induced Superoxide Anion is Affected by pex7-1 Mutation

Considering that the trend of the levels of RNS, namely NO and ONOO^−^, are similar between wt and mutant even in the presence of Cd, the levels of the superoxide anion were evaluated in the roots of wt and *pex7-1* exposed or not to Cd through NBT staining. In the wt, both Cd concentrations, but mainly the higher one, caused an increased NBT signal in PRs and LRs, but mainly in the latter ones (Figure 7a–f, Appendix A). The superoxide radical signal was absent in the apices of the wt PRs and LRs and weak in the vascular system in the Control treatment (Figure 7a,b and inset in a). The signal highly increased in Cd presence, and in a concentration dependent manner, in both wt PR and LRs, extending also to the apex of the latter ones (Figure 7c–f). The localization of the signal did not change in the mutant (Appendix A). However, it was weaker than the wt, both in the PRs and LRs and in the absence and presence of Cd (Figure 7g–l). Results show that the mutation affects the levels of this ROS.

### 3.7. pex7-1 Mutation Delays H_2_O_2_ Scavenging in Cd-Exposed Roots

The levels of H_2_O_2_ were observed three, four, and ten days after germination (DAG) in the PRs and LRs of wt and *pex7-1* exposed or not to Cd through the DCFH-DA probe. The quantification of the H_2_O_2_ fluorescence signal was carried out on the root apices and elongation zones of the wt and *pex7-1* PRs at the same time points. After ten days of Cd treatment, a significant (*p* < 0.01) reduction of the fluorescence signal, related to H_2_O_2_ was observed in the PRs and LRs of wt exposed to 30 and 60 µM Cd (Figure 8a–f,m).

The reduction of the level of the signal was further confirmed by the DAB histochemical analysis, which also evidenced the absence of H_2_O_2_ in the PRs and LRs apices of the wt (Figure 9a–f).

In the mutant roots, with both the analytical methods used, the H_2_O_2_ levels also decreased, and mainly in the presence of the higher Cd concentration (Figure 8g–l,n and Figure 9g–l). However, H_2_O_2_ levels in *pex7-1* PRs exposed to 60 µM Cd remained slightly higher than those observed in wt PRs under the same treatment (Figure 8k,m–n and Figure 9e,k).

To check if there was a change in the PR H_2_O_2_ levels over time, the evaluation of its levels was carried out at several days after germination before day ten. After three days of germination, similar levels of H_2_O_2_ were observed in the wt roots independently from Cd exposure (Figure 8m). On day four, in the wt, the H_2_O_2_ level slightly decreased in the Control, but then sharply decreased up to day ten (Figure 8m). By contrast, the signal of this ROS strongly and significantly decreased in Cd presence already at day four, and in a concentration dependent manner, but remained quite unchanged during the following days (Figure 8m). On day three, the levels of H_2_O_2_ in the mutant were already lower than those of the wt under the Control treatment, whereas they were higher than the wt in the presence of 30 µM Cd (Figure 8m,n). Interestingly, in the Control treatment, the levels of H_2_O_2_ in the mutant did not change within the ten days of culture (Figure 8n). With Cd, the signal of this ROS decreased significantly (*p* < 0.05) in comparison with day three, and similarly for both concentrations. Later, the signal further decreased, but only with 60 µM Cd (Figure 8n).

Root CAT activity was evaluated in the wt and *pex7-1* seedlings after ten days of Cd exposition. The enzyme activity was significantly (*p* < 0.01) reduced by 40% in the *pex7-1* roots in comparison with the wt already in the Control treatment. After the exposure to 30 and 60 µM Cd treatments, there was a reduction of 42.5 and 61.1% in the enzyme activity in the roots of wt with respect to the Control, while in *pex7-1* the percentage of reduction was higher, i.e., of 53.8 and 77.9%, respectively (Figure 10).

Overall, these results highlight a role of PEX7 in the root CAT activity. Indeed, although Cd exposure causes a reduction of the enzyme activity in both genotypes, *pex7-1* showed impaired CAT activity also in Control conditions, and a higher enzyme sensitivity to the heavy metal treatments compared to the wt (Figure 10). In this regard, the differences between the two genotypes in the variation of the root H_2_O_2_ levels detected over time (Figure 8m,n) might be related to their different CAT activity.

## 4. Discussion

Results show that the PEX7 peroxisomal importer is positively involved in Cd translocation from roots to shoots, because the capability to translocate Cd from the root to the shoot is reduced in the mutant. The peroxisomal root signal is enhanced in response to the pollutant, and mutation-related defects in peroxisomes result into an increase in sensitivity to the heavy metal toxicity, because the root protoplasts of the mutant show a progressive decrease in the peroxisomal signal in the presence of Cd. This suggests that a correct import by PEX7 of PTS2 proteins into the peroxisomal matrix ameliorates the root reaction to the pollutant. The exposure to Cd alters the morphology of the root system by reducing PR length and LR formation and elongation and by enhancing the auxin content of shoots and roots. *pex7-1* mutation reduces the IBA-to-IAA conversion, contributing to alter auxin balance and the hormonal response to Cd. PEX7 activity is involved in changing ROS levels and is required for an optimal CAT scavenging activity. Altogether, results demonstrate that well-functioning peroxisomes are indispensable to the root system for reacting to Cd, and that a correct import of PTS2-proteins by PEX7 into the matrix is essential for a proper ROS-mediated signal transduction pathway induced by the pollutant and for an optimal ROS-scavenging activity of CAT.

### 4.1. PEX7 Has a Morphogenic Role in the Root System in Response to Cd

In the non-hyperaccumulator species, such as *Arabidopsis thaliana*, Cd content in the roots is greater than in the aboveground plant tissues [72], because only a small proportion of Cd is transported to the aboveground organs [73], as also confirmed by the present results. It is known that Cd-stress induces peroxisomal senescence in tomato leaves, activating numerous enzymes, as well as senescence-associated peroxisomal peptidases [74]. In *Arabidopsis thaliana*, Cd induces oxidative stress affecting cellular redox homeostasis, with peroxisomal activity involved [37]. Interestingly, it is here shown that Cd cellular response in *Arabidopsis thaliana* roots involves the activity of the AtPEX7 peroxisomal importer, and that this is important for Cd translocation to the shoot. Moreover, we show that the Cd-induced root peroxisomal signal is reduced in *pex7-1*, revealing an increased sensitivity to the heavy metal toxicity of the mutant, possibly deriving from its defective peroxisomes. This highlights that these organelles are determinant for Cd-response of root cells, and it is possible that their disfunction in *pex7-1* involves more proteins than the PTS2 ones requiring the PEX7 importer. In fact, in *Arabidopsis thaliana*, the binding of PEX7 to PTS2 proteins requires PEX5 as a co-receptor [14], and *pex7-1* mutation results in reduced protein levels of both PEX7 and PEX5, as well as in a reduced import of PTS1 and PTS2 cargoes [75]. However, PEX proteins other than PEX5 might be also involved. For example, the receptor docking PEX13, which is known to be crucial for cell survival [6], interact with PEX7 but not with PEX5 [76].

Present data show that PEX7 has an important, and totally new, morphogenic role in the root system in response to Cd. In fact, in the presence of Cd, *pex7-1* mutation results in a reduced LR elongation, with the main part of LRs remaining blocked at the primordium stage, whereas the PR response to Cd does not change significantly from the wt. The lower amount of elongated LRs in *pex7-1* with respect to the wt probably depends on the reduced IBA-to-IAA conversion pathway, which could characterize this line, as assumed by Woodward and Bartel [14]. Indeed, the IBA-derived IAA has pivotal roles in various aspects of root development and in LR development in particular [77]. Cadmium toxicity is known to cause a reduction in PR elongation, which is coupled with an accelerated differentiation of the primary tissues [25,27], whereas the effects on LR elongation have been less studied. However, in the presence of Cd, different results were obtained for LR induction, because in some cases, and depending on the culture conditions, it was enhanced [25], but in others it was reduced [78] and presented results. The absence of differences between *pex7-1* and the wt PRs demonstrates that PEX7 does not affect the developmental response of the PR to the pollutant. In accordance, *pex7-1* responds normally to exogenous IAA in inhibiting PR elongation [14], a process known to be under IAA control [79], and unaffected by the mutation (present results). The LRs, more than PR, are involved in adaptive and acclimation strategies of plants to adverse environments [25]. However, data about the formation/development of LRs in the presence of Cd are contrasting [25,78,79]. In partial accordance with the literature data, present results show a reduction of elongated LRs caused by the pollutant, because most of them remain blocked at the LRP stage, even if this occurs with the lower Cd concentration only. Moreover, our results also show that the formation of functional LRs was further reduced by the mutation in the presence of the pollutant, and independently of its concentration. This suggests that PEX7 is involved in counteracting this root system anomaly caused by Cd, and that it probably collaborates in the activation of PR pericycle cells competent to LRP formation, and in the further LR elongation, to compensate Cd negative effects.

### 4.2. Endogenous IAA and IBA Protect the Root system from Cd Stress

The auxin IAA, and its precursor IBA, may exert a positive role in ameliorating the root system response to Cd stress, as recently demonstrated in rice [29]. In fact, in the latter plant, exogenous treatments with IAA, but mainly with IBA, combined with Cd, mitigate the pollutant effects on the roots [29]. However less information is present in the literature on the endogenous levels of the two auxins. Present data show that the endogenous IAA is higher in the presence of Cd that in its absence. This means that, under Cd-related stress conditions, the hormone is not used to build the LRPs and favor their elongation in LRs, but its levels remain high to counteract Cd toxicity in another way. The endogenous levels of IAA are lower in *pex7-1* than the wt, but are higher in Cd presence than in Cd absence, and this occurs in the mutant as in the wt, confirming their similar IAA response, with/without Cd. It is known that *pex7-1* is insensitive to exogenous IBA [14]. Exogenous IBA is a root inducer better that IAA in a lot of plants and in vitro systems [80]. To function as root inducer, IBA must be converted into IAA, with this conversion occurring in the peroxysomes [81,82]. Present data show that *pex7-1* exhibits dysfunctional peroxysomes, and that the endogenous levels of IBA remain high in the mutant by a reduced/impossible conversion into IAA, in accordance with a literature hypothesis [14]. This mainly occurs in Cd presence, supporting the alteration in lateral rooting response here observed. The levels of IBA increase also in the shoot of the *pex* mutant. Thus, it is possible that also IBA contributes to improve the plant defense to the pollutant toxicity, with its levels remaining as high as those of IAA, because it is not being used for the LRs. The high levels of the two endogenous auxins might result in their reduced use for the biosynthesis of stress- and auxin-related hormones, such as ethylene and jasmonates, whose roles as root inducers have been demonstrated [22,24,83]. In accordance with this, increased levels of the two auxins have been detected in different plant species after the exposure to different stresses, such as drought [84], osmotic [85], or toxic metals [86]. Furthermore, the hormone cytokinin has also recently been shown to play a role in the defense responses of plants from Cd by activating the antioxidant system of plant cells which, as mentioned above, is mainly regulated by peroxisomal metabolism [87]. Taken together, the results show that the two main plant auxins, but also other plant hormones, may be involved in a protection strategy against the pollutant uncoupled with their developmental role as root inducers.

### 4.3. pex7-1 Mutation Negatively Affects Cellular Homeostasis of ROS, But Does Not Affect RNS Homeostasis

Heavy metals, and therefore also Cd, increase RNS and ROS synthesis in plant cells by inducing modifications of the peroxisomal oxidative metabolism [88] to counteract or bear the cellular damages. Our results demonstrate that NO and peroxynitrite levels due to Cd treatments show the same trend in the wt and *pex7-1* roots, with the higher Cd concentration inducing a higher level of these two RNS in both genotypes. It is known that Cd can either reduce or increase NO levels in *Arabidopsis thaliana* roots [37,89] depending on Cd levels, duration of exposure, and culture conditions [90]. It is also known that defined NO levels can alleviate heavy metal toxicity acting as signal molecules by activating the cellular antioxidant system, and thus alerting the cells to respond to the stress conditions [91]. The high levels of NO in *pex7-1* roots after exposure to 60 µM Cd, as well as in wt roots, suggest that the mutation does not interfere with the ability of cells to synthesize NO despite the defect in the peroxisome metabolism. The contrast of this result with previous published data [92], could be explained by the high NO diffusibility and reactivity, which make the levels of this signal molecule highly variable depending on the growth conditions and the plant organ/tissue analyzed. Otherwise, Cd might trigger a NO synthesis pathway, alternative to the peroxisomal one, as a compensation mechanism. The latter hypothesis can be also supported by the reduced peroxisomal signal in root protoplasts after 60 µM Cd treatment here observed.

Nitric oxide reacts quickly with O_2_^●−^ to form peroxynitrite (ONOO^−^), a powerful oxidant, which induces oxidation and nitration of key molecules [93]. It has been demonstrated that in *Arabidopsis thaliana,* ONOO^−^ is produced in the peroxisomes and that it is overproduced under Cd stress [30]. Similarly, in our experiments, both in wt and in *pex7-1* roots, the peroxynitrite levels increase significantly in the presence of the higher Cd concentration. These results demonstrate that the mutation in *PEX7* does not alter the biosynthetic pathway of the ONOO^−^. Altogether, the results concerning NO and ONOO^−^ trends show that the *Arabidopsis thaliana* peroxin PEX7 has a minor role in the modulation of cellular RNS, at least under our cultural conditions.

On the contrary, we demonstrate that ROS, in particular superoxide anion and hydrogen peroxide production, is affected by *pex7-1* mutation. The signal of the superoxide anion detected by nitro blue tetrazolium, increases significantly in the roots of the wt in parallel with the concentration of Cd used, in accordance with other reports [80]. On the contrary, a similar increase in the O_2_^●−^ signal was less evident in the mutant roots. We also show that already at three days after germination, hydrogen peroxide levels are higher in the wt than in the mutant, both in the Control treatment and in the presence of Cd. These levels drop quickly, especially in the roots exposed to Cd and, in fact, after four days from germination they reach low values, which then remain constant up to the end of the culture period. This trend is in line with a correct modulation of H_2_O_2_ levels implemented by cells that control the oxidative metabolism, also because of a good functioning of their peroxisomes. These results are in accordance with what has been reported for *Glycine max* roots and for other plant species exposed to Cd [94,95]. In the mutant roots not exposed to Cd, a late (day four) and lower peak in H_2_O_2_ level was observed in comparison with the wt, suggesting a dysfunction linked with the altered peroxisomes. In fact, it is known that peroxisomes can synthesize H_2_O_2_ from various metabolic pathways, including β-oxidation [96] and *pex7-1* mutant, which has a defective β-oxidation. Thus, the reduced levels of H_2_O_2_ in the mutant roots could be also related to reduced β-oxidation. It is possible that different peroxisomal pathways of H_2_O_2_ production are activated in relation to the plant developmental stages [96] and to stress conditions sustaining our results.

Despite the very low affinity of CAT to H_2_O_2_, the high abundance of this enzyme compensates for this, making it the most important H_2_O_2_ regulator in plant cells [97]. Indeed, CAT comprises as much as 10–25% of the total peroxisomal proteins [98] and its import in the peroxisomal matrix is dependent on the functionality of PTS targeting signals, mainly PTS1 [99,100]. CAT activity is inhibited by Cd in rice roots [101], such as in *Arabidopsis thaliana* roots (present results). In our experiments, CAT activity is significantly reduced in the mutant, in comparison with the wt, and the Cd presence further decreases it in both genotypes. Moreover, our results allow us to hypothesize that the altered ROS levels detected in the mutant with respect to the wt could affect other components of the peroxisomal protein import machinery related to CAT import, as also demonstrated in mammalian cells [102]. In addition, as already stated, *pex7-1* mutant could show defects in PTS1 import other than in PTS2 [75], probably leading to a lower CAT content with respect to the wt.

However, PTS2 proteins might also be involved in regulating CAT activity in *Arabidopsis thaliana* roots mainly in the presence of Cd stress, because the altered import of peroxisomal proteins due to the lack of their cytosolic receptor PEX7 might result into very low levels of CAT activity, based on present data. In fact, our results show that the combined effects of *PEX7* mutation and Cd reduce activity of CAT, causing higher levels of H_2_O_2_ in the mutant roots.

## 5. Conclusions

The overall composition of peroxisomal proteins determines peroxisome functions and makes the organelle able to attend in counteracting the toxicity of heavy metals such as Cd. This imply that peroxisomal matrix protein import, because of the functionality of PTS targeting signals (PTS1 and PTS2) recognized by PEX receptors, is essential to contrast the heavy metal damages. Our results suggest that PEX7, and consequently PTS2-targeting signal, are involved in the control of Cd toxicity and this is mainly achieved by controlling ROS metabolism and affecting auxin levels. However, further analyses using mutant lines for other *pex7* alleles must be carried out to confirm these results. A comprehensive understanding of the mechanisms that model/useful plants use at the cellular level by the activity of organelles such as peroxisomes and their proteome in defending themselves against metal toxicity will allow phytoremediation technology to take a step forward in the treatment of heavy metal-polluted soils.

## Figures and Tables

**Figure 1 antioxidants-10-01494-f001:**
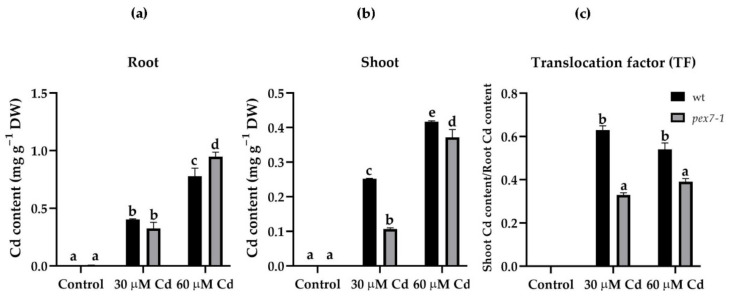
Mean values (±SE) from three technical replicates of cadmium (Cd) accumulation (mg g^−1^ DW) through ICP-OES in roots (**a**) and shoots (**b**) of wt and *pex7-1* seedlings not exposed (Control) or exposed to 30 µM CdSO_4_ (30 µM Cd) or 60 µM CdSO_4_ (60 µM Cd) and of the translocation factor (**c**) of wt and *pex7-1*. Columns labelled with different letters show statistical differences for at least *p* < 0.05 among treatments within the same genotype and between genotypes within the same treatment. Columns labelled with the same letter are not statistically different (*p* > 0.05). Data are from the first biological replicate.

**Figure 2 antioxidants-10-01494-f002:**
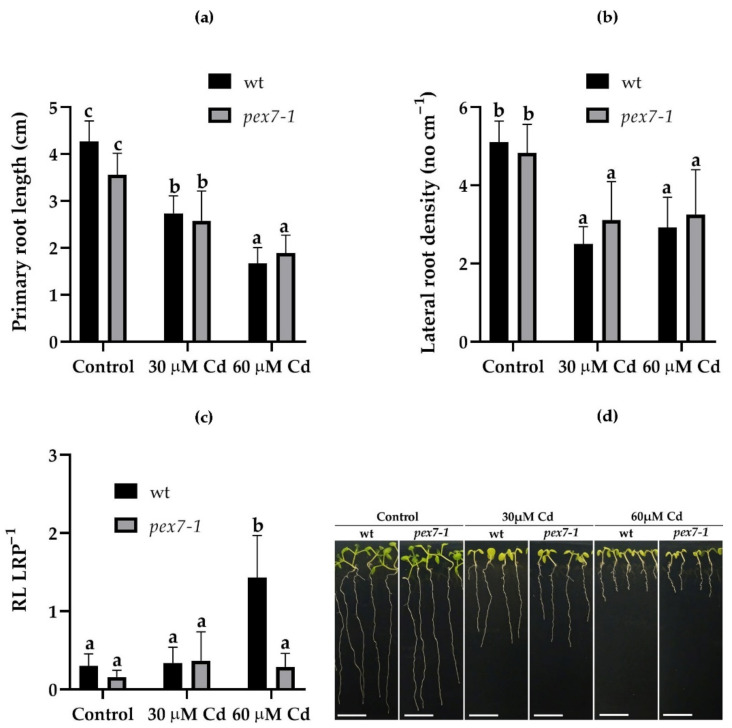
Mean values (±SE) (from three biological replicates) of the primary root length (**a**), lateral root density (**b**), of the ratio between the number of mature lateral roots (LR) and the lateral root primordia (LRP) (**c**) and macroscopic images (**d**) of wt and *pex7-1* seedlings not exposed (Control) or exposed to 30 µM CdSO_4_ (30 µM Cd) or 60 µM CdSO_4_ (60 µM Cd). Columns labelled with different letters show statistical differences for at least *p* < 0.05 among treatments within the same genotype and between genotypes within the same treatment. Columns labelled with the same letter or with no letter are not statistically different (*p* > 0.05). *n* = 30. Bars = 1 cm.

**Figure 3 antioxidants-10-01494-f003:**
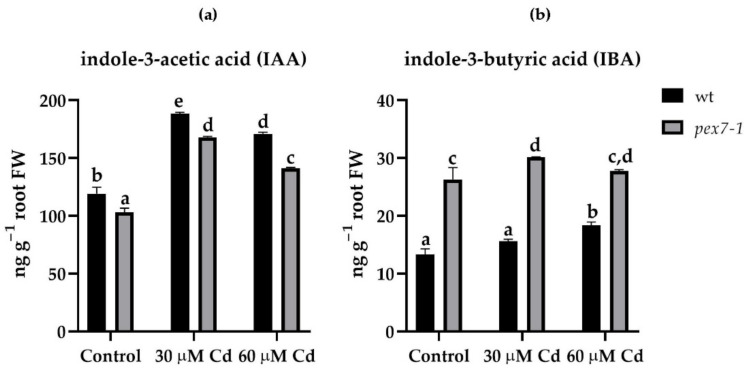
Mean values (±SE) from three technical replicates of IAA (**a**) and IBA (**b**) content (ng g^−1^ FW) in roots of wt and *pex7-1* seedlings not exposed (Control) or exposed to 30 µM CdSO_4_ (30 µM Cd) or 60 µM CdSO_4_ (60 µM Cd). Columns labelled with different letters show statistical differences for at least *p* < 0.05 among treatments within the same genotype and between genotypes within the same treatment. Columns labelled with the same letter are not statistically different (*p* > 0.05). Data from the first biological replicate.

**Figure 4 antioxidants-10-01494-f004:**
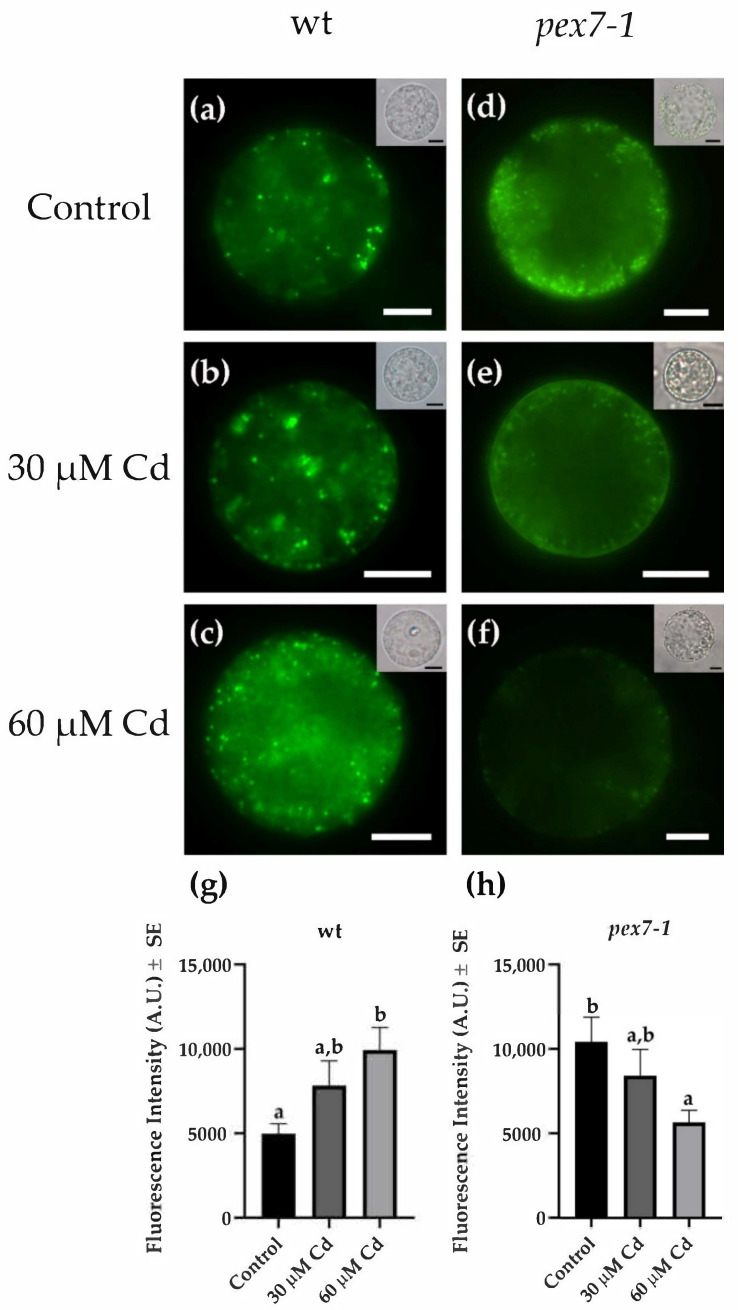
N-Bodipy epifluorescence signal in wt (**a**–**c**) and *pex7-1* (**d**–**f**) root protoplasts from seedlings grown in the absence ((**a**,**d**), Control) or in the presence of 30 µM CdSO_4_ (30 µM Cd) (**b**,**e**) or 60 µM CdSO_4_ (60 µM Cd) (**c**,**f**). Insets in (**a**–**f**) show the same field under white light. (**g**,**h**) Mean values (±SE) from three biological replicates of N-Bodipy fluorescence intensity in root protoplasts of the same size (~200 µm^2^) measured using ImageJ software (version 1.53c, Wayne Rasband, National Institutes of Health, Bethesda, USA) and expressed in arbitrary units (A.U.s; from 0 to 255). Columns labelled with different letters among treatments show statistical differences for at least *p* < 0.05 level. Columns labelled with the same letter are not statistically different (*p* > 0.05). Bars = 7 µm (**a**–**c**) and 10 µm (**d**–**f**). *n* = 30.

**Figure 5 antioxidants-10-01494-f005:**
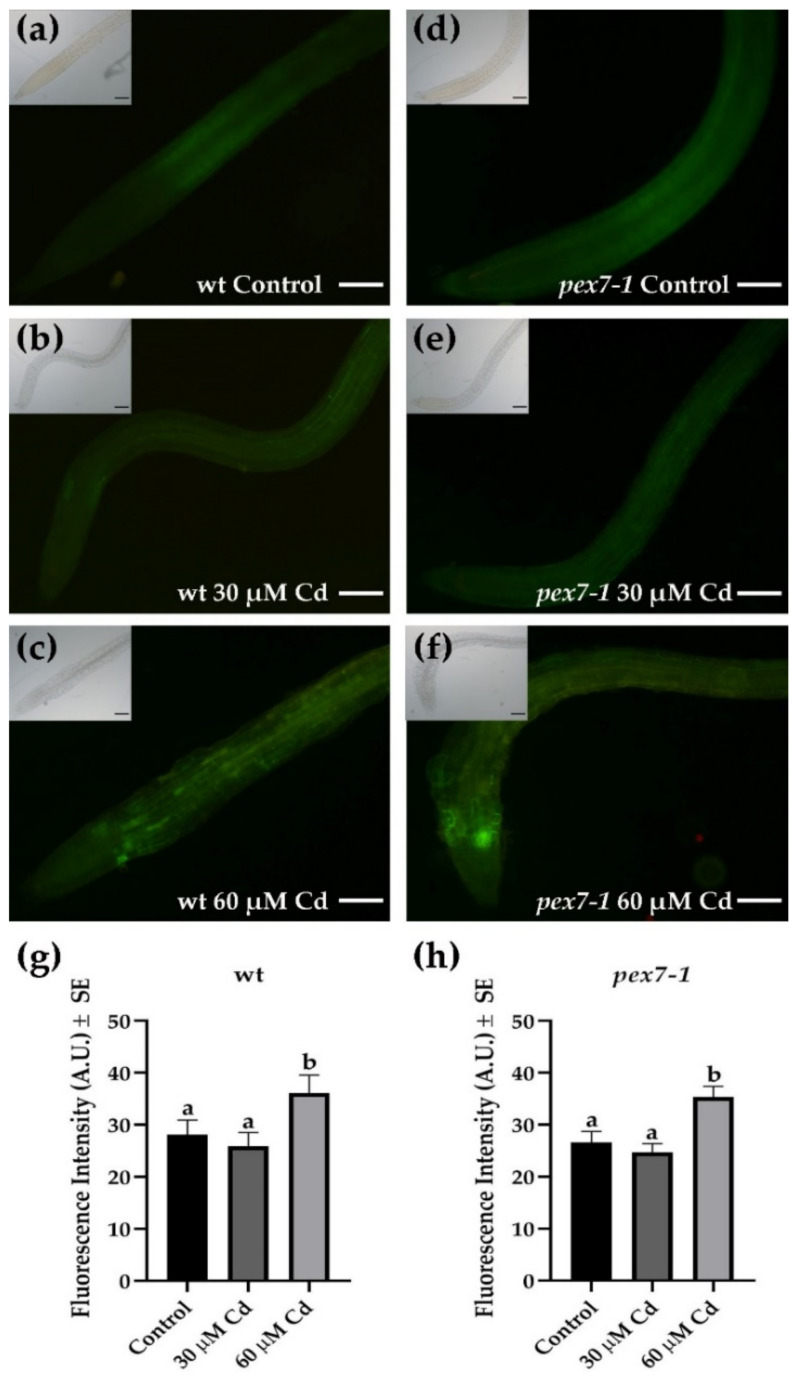
DAF-FM DA epifluorescence analysis showing nitric oxide (NO) signal in the wt (**a**–**c**) and *pex7-1* (**d**–**f**) roots from seedlings grown in the absence ((**a**,**d**), Control) or in the presence of 30 µM CdSO_4_ (30 µM Cd) (**b**,**e**) or 60 µM CdSO_4_ (60 µM Cd) (**c**,**f**). Insets in (**a**–**f**) show the same field under white light. (**g**,**h**) mean values (±SE) from three biological replicates of DAF-FM DA fluorescence intensity measured using ImageJ software (version 1.53c, Wayne Rasband, National Institutes of Health, Bethesda, USA) and expressed in arbitrary units (A.U.s; from 0 to 255). Columns labelled with different letters among treatments show statistical differences for at least *p* < 0.05 level. Columns labelled with the same letter are not statistically different (*p* > 0.05). Bars = 100 µm. *n* = 30.

**Figure 6 antioxidants-10-01494-f006:**
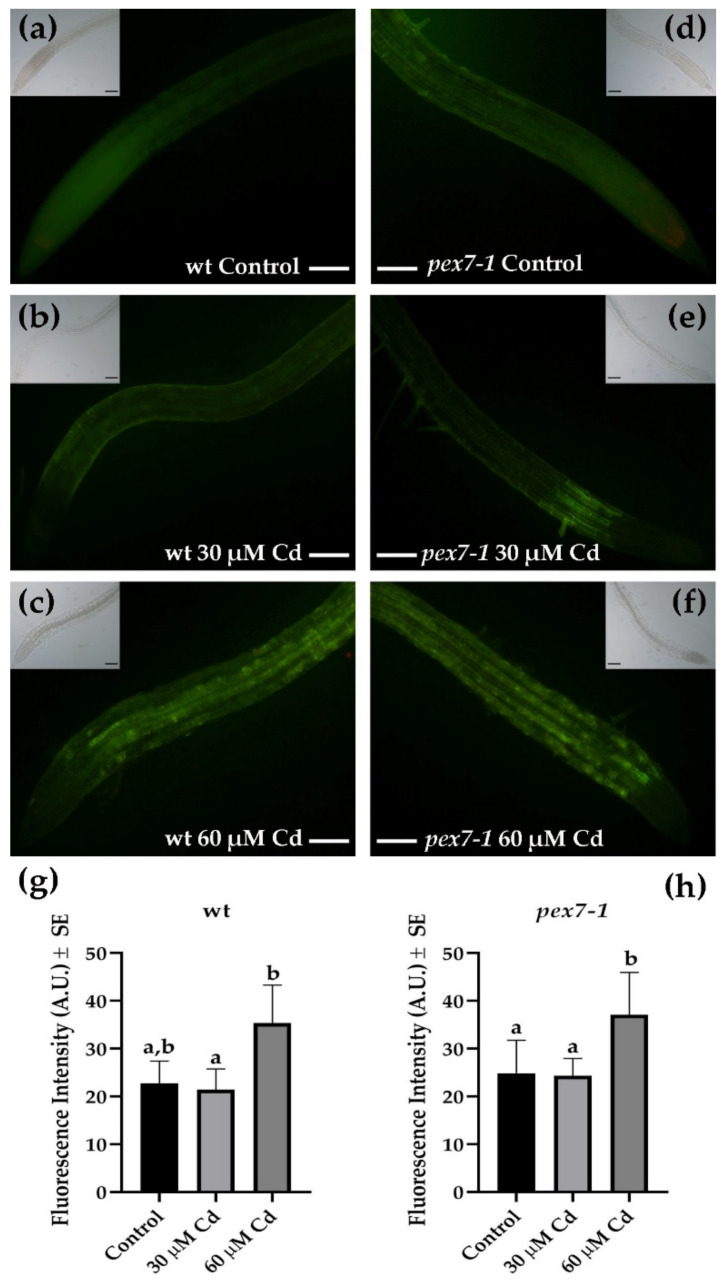
APF epifluorescence analysis showing peroxynitrite (ONOO^−^) signal in the wt (**a**–**c**) and *pex7-1* (**d**–**f**) roots from seedlings grown in the absence ((**a**,**d**) Control) or in the presence of 30 µM CdSO_4_ (30 µM Cd) (**b**,**e**) or 60 µM CdSO_4_ (60 µM Cd) (**c**,**f**). Insets in (**a**–**f**) show the same field under white light. (**g**,**h**) Mean values (±SE) from three biological replicates of APF fluorescence intensity measured using ImageJ software (version 1.53c, Wayne Rasband, National Institutes of Health, Bethesda, USA) and expressed in arbitrary units (A.U.s; from 0 to 255). Columns labelled with different letters among treatments show statistical differences for at least *p* < 0.05 level. Columns labelled with the same letter are not statistically different (*p* > 0.05). Bars = 100 µm. *n* = 30.

**Figure 7 antioxidants-10-01494-f007:**
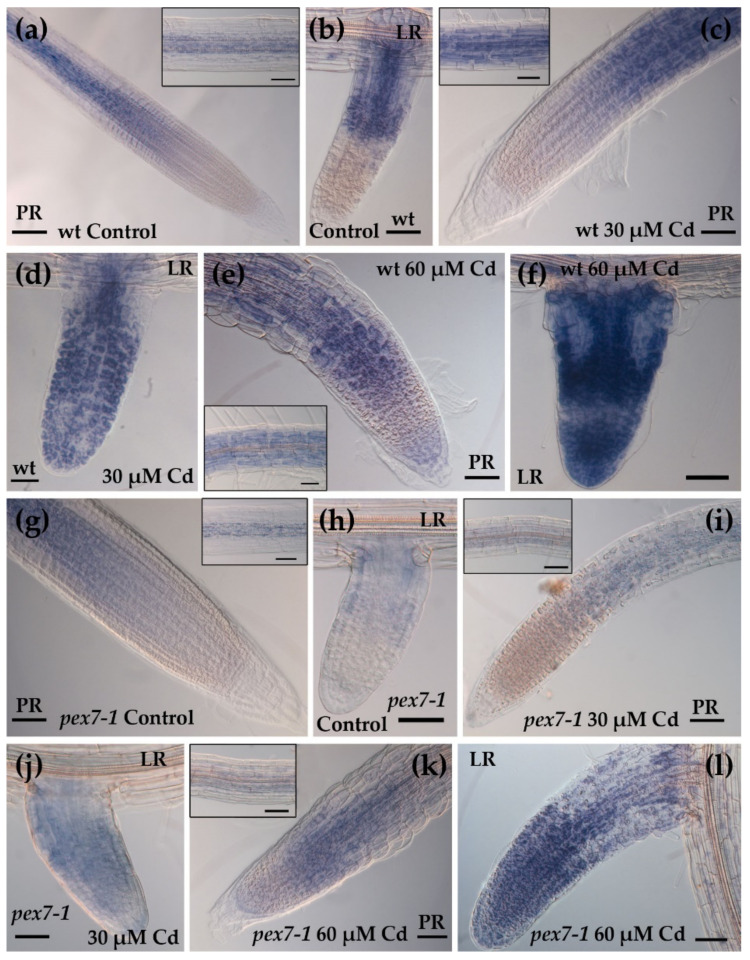
NBT histochemical analysis showing O_2_^●−^staining in the primary (PR) and lateral (LR) roots of wt (**a**–**f**) and *pex7-1* (**g**–**l**) seedlings grown in the absence ((**a**,**b**,**g**,**h**), Control) or in the presence of 30 µM CdSO_4_ (30 µM Cd) (**c**,**d**,**i**,**j**) or 60 µM CdSO_4_ (60 µM Cd) (**e**,**f**,**k**,**l**). The differentiation zone with xylary cells is shown in the insets. Images came from the three biological replicates. Bars = 100 µm (**a**), 50 µm ((**b**,**c**, **f**–**l**), insets), 25 µm (**d**,**e**). *n* = 30.

**Figure 8 antioxidants-10-01494-f008:**
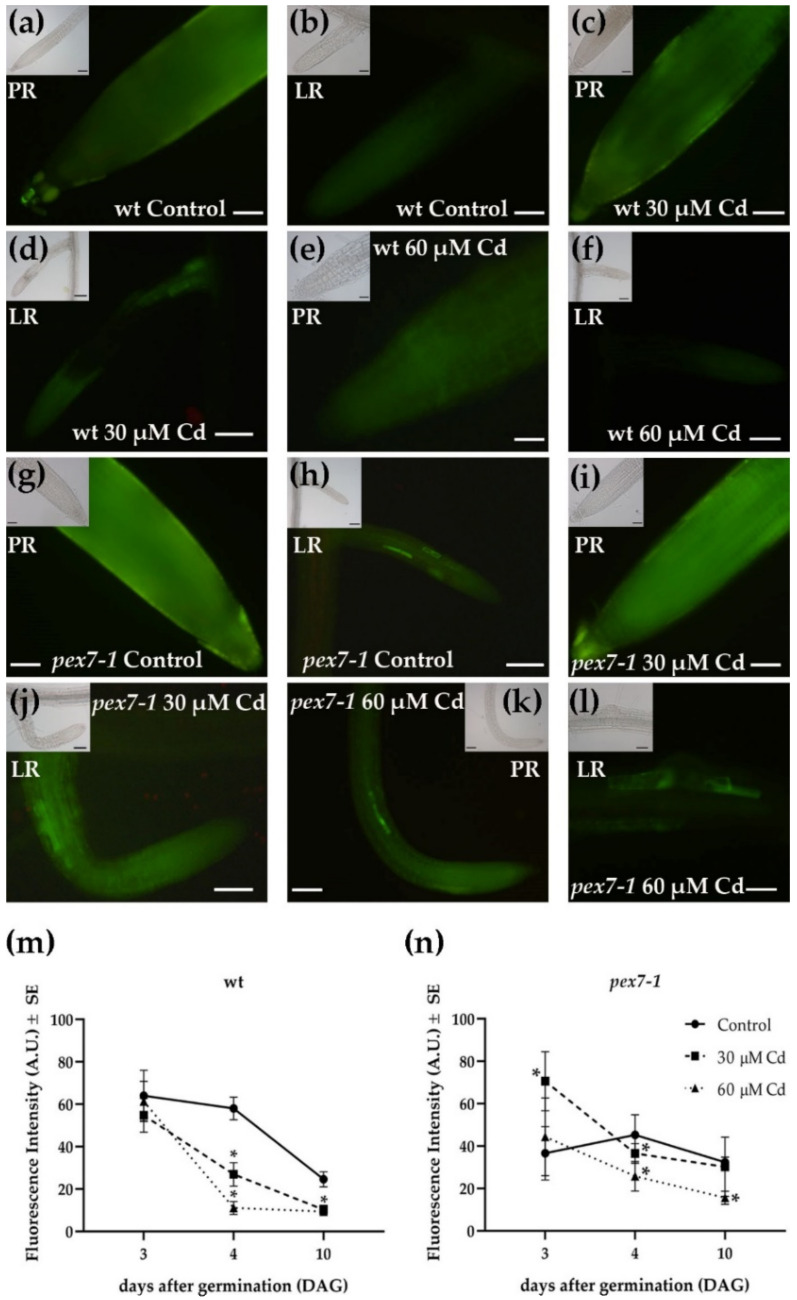
DCFH-DA epifluorescence analysis showing H_2_O_2_ signal in the primary (PR) and lateral (LR) roots of wt (**a**–**f**) and *pex7-1* seedlings (**g**–**l**) grown in the absence ((**a**,**b**,**g**,**h**), Control) or in the presence of 30 µM CdSO_4_ (30 µM Cd) (**c**,**d**,**i**,**j**) or 60 µM CdSO_4_ (60 µM Cd) (**e**,**f**,**k**,**l**). Insets in (**a**–**l**) show the same field under white light. (**m**,**n**) Mean values (±SE) from three biological replicates of DCFH-DA fluorescence intensity at different days after germination (DAG) measured in the PR apices and elongation zones using ImageJ software (version 1.53c, Wayne Rasband, National Institutes of Health, Bethesda, USA) and expressed in arbitrary units (A.U.s; from 0 to 255). Asterisks show statistical differences for at least *p* < 0.05 level with respect to the Control within the same day. Bars = 100 µm (**d**,**f**,**h**,**j**,**k**) and insets in (**a**,**d**,**f**,**h**,**j**,**k**), 50 µm (**a**–**c**,**e**,**g**,**i**,**l**) insets in (**b**,**c**,**e**,**g**,**i**,**l**). *n* = 30.

**Figure 9 antioxidants-10-01494-f009:**
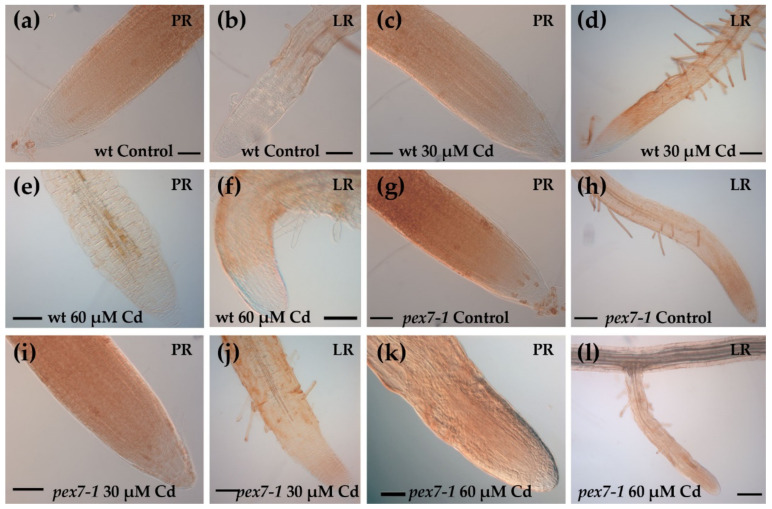
DAB histochemical analysis showing H_2_O_2_ staining in the primary (PR) and lateral (LR) roots of the wt (**a**–**f**) and *pex7-1* (**g**–**l**) seedlings grown in the absence ((**a**,**b**,**g**,**h**), Control) or in the presence of 30 µM CdSO_4_ (30 µM Cd) (**c**,**d**,**i**,**j**) or 60 µM CdSO_4_ (60 µM Cd) (**e**,**f**,**k**,**l**). Images came from the three biological replicates. Bars = 100 µm (**d**–**f**,**h**,**j**–**l**), 50 µm (**a**–**c**,**g**,**i**). *n* = 30.

**Figure 10 antioxidants-10-01494-f010:**
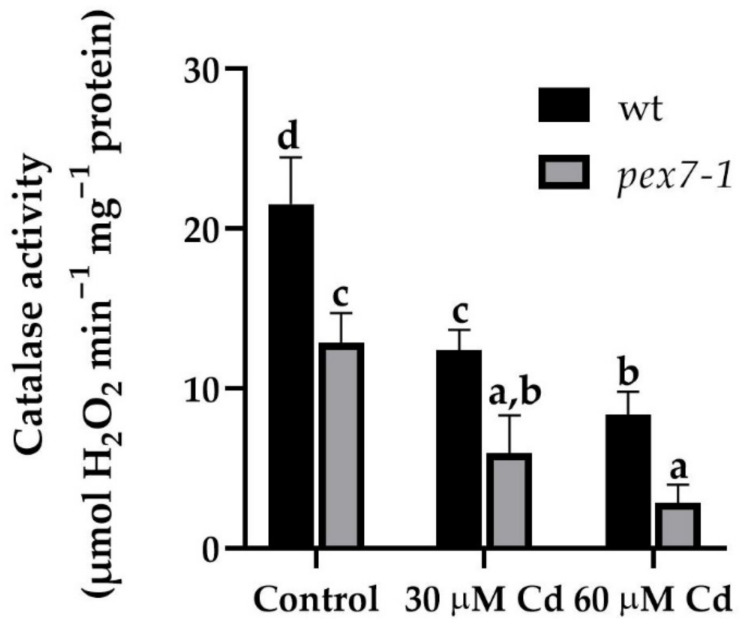
Catalase (CAT) activity, expressed as μmol H_2_O_2_ min^−1^ mg^−1^ protein in the root system of wt and *pex7-1* seedlings grown in the absence (Control) or in the presence of 30 µM CdSO_4_ (30 µM Cd) or 60 µM CdSO_4_ (60 µM Cd). Columns labelled with different letters show statistical differences for at least *p* < 0.05 among treatments within the same genotype and between genotypes within the same treatment. Columns labelled with the same letter are not statistically different (*p* > 0.05). Data are means (±SE) of three technical replicates from the first biological replicate.

## Data Availability

The data presented in this study are available in the article and Appendix A.

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
