# Peer review of "Peroxisomal PEX7 Receptor Affects Cadmium-Induced ROS and Auxin Homeostasis in Arabidopsis Root System"

_antioxidants, 2021, doi:10.3390/antiox10091494_

Round 1

Reviewer 1 Report

The manuscript presented by Piacentini and colleagues study the physiological and biochemical relationships between the PEX7 peroxisomal receptor and the plant response to cadmium toxicity using Arabidopsis thaliana as model. The experiments seem to be well conducted, the data are overall clearly presented and the manuscript is well written. However, there are several issues listed below that need to be addressed (in particular the statistics). It is also important that the authors remain careful in their conclusions since only one pex7 mutant allele was use and that there was no complemented pex7-1 mutant line analysed.

Major points

- Lines 291 to 312 (and throughout the document). The authors should use “wt” and not “Col-0”.

- Lines 307 to 312. The authors should include the translocation factor (TF) into Figure 1 as panel (c).

- Lines 314 to 315. Figure 2, the panel dealing with PR length should be panel (a) and not an inset of the panel (a). Thus the actual panel (a) should be panel (b) etc. In addition, the authors should provide picture of whole seedlings as well as of the whole root system. This would be very useful for the reader to fully assess the combined effects of the pex7 mutation and the Cd treatments.

- Lines 317 to 319. Based on Figure 2a results, it is unlikely that the lateral root density is different between the WT and the mutant. Since there are 3 biological repeat per columns, it might be that there is some problems with the statistics given that an ANOVA with Tuckey test was used (there should be no difference between the genotype in a given condition).

- Lines 320 to 322. Figure 2b, the statistical results are not fully provided.

- Lines 340 to 343. Figure S1, the pex7 mutant does not show an increased accumulation of IBA in shoots when compared to the wt in the presence of Cd. In addition, in Figure S1, the statistical results are not fully provided. Based on these data, it seems that the differences in IBA conversion into IAA between both genotypes are robust in roots but are not supported in shoots. In addition, it is not clear based on the data that there is more IBA accumulation in the pex7 mutant when grown under 30 mM or 60 mM Cd than in the control condition. The authors should temper down their conclusion.

- Lines 363 to 365. The N-BODIPY results do not demonstrate that pex7 mutant is more sensitive to heavy metal toxicity than the wt. The authors should temper down this conclusion. These data suggest that peroxisome defects due to Cd treatment are occurring in the pex7 mutant. Is the N-BODIPY pattern more diffuse in the mutant than in the wt?

- Lines 416 to 418. The authors should indicate that they refer to NBT staining.

- Lines 426 to 427. It would be useful for the readers that the authors provide as supplementary material pictures displaying whole seedlings following NBT staining.

- Lines 435 to 438. Which fluorescence signal was analyzed should be indicated in the main text. In addition, Figure 8 it is not possible to conclude on LR fluorescence since there is no data/picture for the wt in the 30 mM or 60 mM Cd conditions and for the mutant in the control or 60 mM Cd conditions. This should be included in the figure. What is measure in panel (i) and (j) is unclear. Is it the whole fluorescence (i.e. whole seedlings), the whole roots or is it the root tips? This should be clarified. Were the settings used for the microscope the same for all the conditions and genotypes? The same part of the root should be shown for panels (a) to (h) and for the missing pictures. Panel (j), there is no statistically significant difference at 3 days between the control and the 30 mM Cd condition while there is a statistically significant difference at 4 and 10 days. This is surprising since the points are closer at 4 and 10 days between the samples than at 3 days. Last, if panel (i) and (j) the asterix are to highlight differences between the Cd conditions and the control, the asterix should be on the Cd conditions and not the control ones.

- Lines 446 to 448. If the data for the PR seems quite convincing, the results for the LR are less clear. It seems that for the LR, there is no difference, in term of DAB staining, between all the conditions.

- Lines 472 to 477. The statistics provided Figure 10 are not complete.

Minor points

- Line 15. To what stands for PEX should be stated when it first appears.

- Line 178. What is described Appendix A should be moved to the Materials and Methods section.

- Line 273. What the authors mean by “5x104 mM Tris-HCl”? Is it 50 mM Tris-HCl? This is also true lines 279 and 280.

- Lines 416 to 417. It is described paragraph 3.5 that there is no difference between the wt and the mutant for the production and localization of RNS, so why indicate here that there is “ little influence”?

Author Response

Answers to Referee 1

Comment: It is also important that the authors remain careful in their conclusions since only one pex7 mutant allele was use and that there was no complemented pex7-1 mutant line analysed.

Answer: In the revised version of the manuscript the Conclusions were modified in accordance with your comment.

Comment: Lines 291 to 312 (and throughout the document). The authors should use “wt” and not “Col-0”.

Answer: “Col-0” was replaced by “wt” throughout the document and in the Figures.

Comment: Lines 307 to 312. The authors should include the translocation factor (TF) into Figure 1 as panel (c).

Answer: The data of the translocation factor were reported in the new Figure 1c.

Comment: - Lines 314 to 315. Figure 2, the panel dealing with PR length should be panel (a) and not an inset of the panel (a). Thus the actual panel (a) should be panel (b) etc. In addition, the authors should provide picture of whole seedlings as well as of the whole root system. This would be very useful for the reader to fully assess the combined effects of the pex7 mutation and the Cd treatments.

Answer: the mean value of PR length was reported in the new Figure 2a. Moreover, we added a picture as the new Figure 2c showing whole seedlings of wt and pex7-1 treated or not with CdSO4, including their root systems, as requested.

Comment: Lines 317 to 319. Based on Figure 2a results, it is unlikely that the lateral root density is different between the WT and the mutant. Since there are 3 biological repeat per columns, it might be that there is some problems with the statistics given that an ANOVA with Tuckey test was used (there should be no difference between the genotype in a given condition).

Answer: We agree with the Reviewer, the statistical results were not clear in the old Figure 2a, we modified it in the revised version of Figure 2.

Comment: Lines 320 to 322. Figure 2b, the statistical results are not fully provided.

Answer: The statistical results were fully provided in the new Figure 2.

Comment:  Lines 340 to 343. Figure S1, the pex7 mutant does not show an increased accumulation of IBA in shoots when compared to the wt in the presence of Cd. In addition, in Figure S1, the statistical results are not fully provided. Based on these data, it seems that the differences in IBA conversion into IAA between both genotypes are robust in roots but are not supported in shoots. In addition, it is not clear based on the data that there is more IBA accumulation in the pex7 mutant when grown under 30 mM or 60 mM Cd than in the control condition. The authors should temper down their conclusion.

Answer: The statistical results were fully provided in the new Figure S1.  The last sentence and the conclusion of paragraph 3.3 were changed in accordance with your observations.

Comment: Lines 363 to 365. The N-BODIPY results do not demonstrate that pex7 mutant is more sensitive to heavy metal toxicity than the wt. The authors should temper down this conclusion. These data suggest that peroxisome defects due to Cd treatment are occurring in the pex7 mutant. Is the N-BODIPY pattern more diffuse in the mutant than in the wt?

Answer: We agree with this comment, and the conclusion of paragraph 3.4. was changed accordingly. The N-BODIPY fluorescence pattern is more diffuse in the root protoplasts of mutant than in the wt root protoplasts only when not exposed to Cd. This observation was added on the revised text.

Comment: Lines 416 to 418. The authors should indicate that they refer to NBT staining.

Answer: the use of the NBT staining was indicated in the paragraph 3.6. of the revised version.

Comment: Lines 426 to 427. It would be useful for the readers that the authors provide as supplementary material pictures displaying whole seedlings following NBT staining.

Answer: A new Figure S2 showing longer PRs with their LRs, after NBT staining, was added. The images of the new Figure S2 were performed with Axio Imager M2 microscope motorized on the 3 axes. The Image tiles scanning were carried out with an Axiocam 105 (Zeiss) camera using Zen pro 2.5 (Zeiss) software (see also the new Materials and methods). It was not possible to photograph the entire seedlings owing to limitations in the acquisition and processing of the images due to the software used.

Comment: Lines 435 to 438. Which fluorescence signal was analyzed should be indicated in the main text. In addition, Figure 8 it is not possible to conclude on LR fluorescence since there is no data/picture for the wt in the 30 mM or 60 mM Cd conditions and for the mutant in the control or 60 mM Cd conditions. This should be included in the figure. What is measure in panel (i) and (j) is unclear. Is it the whole fluorescence (i.e. whole seedlings), the whole roots or is it the root tips? This should be clarified. Were the settings used for the microscope the same for all the conditions and genotypes? The same part of the root should be shown for panels (a) to (h) and for the missing pictures. Panel (j), there is no statistically significant difference at 3 days between the control and the 30 mM Cd condition while there is a statistically significant difference at 4 and 10 days. This is surprising since the points are closer at 4 and 10 days between the samples than at 3 days. Last, if panel (i) and (j) the asterix are to highlight differences between the Cd conditions and the control, the asterix should be on the Cd conditions and not the control ones.

Answers: The fluorescence signal analyzed was indicated in the revised version of the manuscript. A new Figure 8 with the images of LRs of wt treated with 30 mM or 60 mM Cd and of LRs of mutant exposed to 60 mM Cd is now provided. In the MM and Result sections of the revised version we specified how and where the fluorescence measurements were performed. The setting of the microscope used for the acquisition of the fluorescence images was kept fixed for the two genotypes, treatments e time points. This was added in the MM section. The statistical results were corrected in the new figure 8m and 8n and in the legend of the new Figure 8.

Comment: Lines 446 to 448. If the data for the PR seems quite convincing, the results for the LR are less clear. It seems that for the LR, there is no difference, in term of DAB staining, between all the conditions.

Answer: We agree, and in the revised version the text related to DAB staining was changed.

Comment: Lines 472 to 477. The statistics provided Figure 10 are not complete.

Answer: The statistical results were fully provided in the new Figure 10.

 Minor points

Comment: Line 15. To what stands for PEX should be stated when it first appears.

Answer: PEX meaning was stated, in the Abstract, in its first appearance.

Comment: Line 178. What is described Appendix A should be moved to the Materials and Methods section.

Answer: the screening procedure of the mutant seedlings was moved to the paragraph 2.2 of the new MM section.

Comment: Line 273. What the authors mean by “5x104 mM Tris-HCl”? Is it 50 mM Tris-HCl? This is also true lines 279 and 280.

Answer: We changed the unit of measurement according to your suggestion

Comment: Lines 416 to 417. It is described paragraph 3.5 that there is no difference between the wt and the mutant for the production and localization of RNS, so why indicate here that there is “ little influence”?

Answer: The beginning of paragraph 3.6 was changed in accordance with your observation.

We are grateful to you and to the referees for helping to improve the manuscript with your comments.

Sincerely yours,

Giuseppina Falasca

Reviewer 2 Report

1.The abstract must include a key message which authors wish to convey to the audience or a statement which they argue.

  1. In the introduction, authors can include the missing information (research gaps and the significance of your research). Why it is required to run such research? What is the economic perspective of this research?
  2. Authors should explain more role of the peroxisomes and the composition of peroxisomal proteins which determines peroxisome functions and makes the organelle able to attend in counteracting the toxicity of heavy metals such as Cd. Authors suggest that peroxisomal matrix protein import, because of the functionality of PTS targeting signals (PTS1 and PTS2) recognized by PEX receptors, is essential to contrast the heavy metal damages.
  3. Authors studied the role of PEX7 in the Arabidopsis thaliana root system exposed to Cd, Cd uptake and translocation, IAA and IBA levels, ROS and RNS levels, and catalase activity were analyzed in pex7-1 mutant
  4. I think this paper is potentially very valuable and represents a good and original study with new results. Results are also presented clearly and correctly. The structure and content of the paper are understandable. In results and discussion, the understanding of the physiological specificity of Cd toxicity is limited, as it is restricted to papers that have a particular view and deliberately ignore alternatives, and does not present a balanced view of the evidence. The authors could discuss more these aspects.
  1. Authors should discuss more alternatives and balance new findings with other approaches. I would have expected a more critical discussion of the results. Please read / use in the discussion following new references: https://doi.org/10.1371/ journal.pone.0249230
  1. The paper requires only MINOR revisions to be acceptable for publication

Author Response

  Answers to Referee 2

Comment: 1. The abstract must include a key message which authors wish to convey to the audience or a statement which they argue.

Answer: In the revised version of the manuscript we modified the last sentence of the abstract in accordance with your comment.

Comment: 2. In the introduction, authors can include the missing information (research gaps and the significance of your research). Why it is required to run such research? What is the economic perspective of this research?

Answer: In the Introduction of the revised manuscript, we have rewritten the purpose of the work by focusing more on the importance of the research. The research was performed using the pex7-1 mutant of the model plant Arabidopsis thaliana which in itself does not have an economic value, but the results provide important information on the processes activated by plants, at the cellular and subcellular levels, for the defense against the toxicity of the heavy metals. This information is useful for improving phytoremediation technology in order to use it more and more in environmental restoration programs with plants of economic value. This concept was added in the Conclusions of the new version of the manuscript.

Comment: Authors should explain more role of the peroxisomes and the composition of peroxisomal proteins which determines peroxisome functions and makes the organelle able to attend in counteracting the toxicity of heavy metals such as Cd. Authors suggest that peroxisomal matrix protein import, because of the functionality of PTS targeting signals (PTS1 and PTS2) recognized by PEX receptors, is essential to contrast the heavy metal damages.

Answer: In the introduction of the revised version we added further information on other peroxisomal proteins and their role in the response to stress conditions.

 .

Comment: I think this paper is potentially very valuable and represents a good and original study with new results. Results are also presented clearly and correctly. The structure and content of the paper are understandable. In results and discussion, the understanding of the physiological specificity of Cd toxicity is limited, as it is restricted to papers that have a particular view and deliberately ignore alternatives, and does not present a balanced view of the evidence. The authors could discuss more these aspects.

Answer: In the new version of the discussion papers with alternative results were added.

Comment: Authors should discuss more alternatives and balance new findings with other approaches. I would have expected a more critical discussion of the results. Please read / use in the discussion following new references: https://doi.org/10.1371/ journal.pone.0249230

Answer: This reference was added in the revised discussion.

We are grateful to you and to the referees for helping to improve the manuscript with your comments.

Sincerely yours,

Giuseppina Falasca